# Disseminated Nocardiosis Caused by *Nocardia farcinica* in Two Puppy Siblings

**DOI:** 10.3390/vetsci10010028

**Published:** 2022-12-30

**Authors:** Flavia Zendri, Peter Richards-Rios, Iuliana Maciuca, Emanuele Ricci, Dorina Timofte

**Affiliations:** Department of Veterinary Anatomy, Physiology and Pathology, Institute of Infection, Veterinary and Ecological Sciences, Leahurst Campus, University of Liverpool, Chester High Road, Neston CH64 7TE, UK

**Keywords:** canine nocardiosis, *Nocardia* spp., *Nocardia farcinica*, disseminated nocardiosis in dogs, central nervous system canine nocardiosis

## Abstract

**Simple Summary:**

Clinical infections caused by bacteria belonging to the genus *Nocardia* spp. are sporadic in humans and animals living in tropical areas; however, recent reports suggest their increasing incidence, including in non-tropical countries. Systemic nocardiosis due to *Nocardia farcinica* may lead to neurologic manifestations in people and similar findings have been reported from infected dogs; therefore, dogs could represent a suitable experimental model for understanding the epidemiology and pathogenetic mechanisms of human infections. Despite *Nocardia* spp. not generally causing outbreaks, these may arise in human hospitals and, as reported with this study for the first time, in canine breeding establishments. Our case report describes the clinical, pathological and microbiological findings associated with fatal systemic infection by *N. farcinica* in two puppies, discussing the current epidemiology in humans and dogs, known pathogenesis and available diagnostics. Overall, our cases displayed pathologic and microbiologic similarities with previously reported human systemic *N. farcinica* infections. In light of the often difficult-to-treat nature of systemic nocardial infections, their neglected and re-emerging status, zoonotic nature and outbreak potential, investigation of canine models may help gain insights of the virulence, clinical-epidemiological features, predisposing factors and identification of *Nocardia* spp. strains isolated from human, animal and environmental sources.

**Abstract:**

Systemic nocardiosis due to *Nocardia farcinica* has not been reported in canine outbreaks. Two 14-week-old female Dogue de Bordeaux siblings presented with fever and severe, acute onset limb lameness; traumatic lesions with evidence of infection were identified over the lame limbs of both dogs. The patients were euthanised owing to lack of therapeutic response and rapid escalation to systemic infection with central nervous system manifestations. The post-mortem changes consisted of multiple disseminated abscesses, mainly affecting the skin and subcutis at the limb traumatic injuries, local and hilar lymph nodes, lung, kidney and brain. Bacterial culture and identification via MALDI-TOF and 16S rRNA sequencing revealed *Nocardia farcinica* from several of these sites in both dogs. Clinical significance of the isolate was supported by cytology of the post-mortem organs’ impression smears showing numerous branching filamentous bacteria associated with inflammation. The organism displayed marked multidrug-resistance. No history of immunosuppression was available, and immunohistochemistry ruled out viral pathogens as canine distemper and parvovirus. *N. farcinica* should be considered as a potential differential cause of sudden lameness and systemic infection in dogs with traumatic skin lesions over the limbs. This is the first reported small-scale outbreak of systemic nocardiosis in dogs due to *N. farcinica*.

## 1. Introduction

Nocardiosis is an acute or chronic, suppurative or pyogranulomatous infection infrequently affecting humans and animals. It is an emerging saprozoonosis of increasing public health significance in both developed and developing countries with relatively high fatality rates [1]. *Nocardia* spp. are ubiquitous saprophytic filamentous Gram-positive organisms common in soil, decaying vegetation, water, air and dust particles. Infection is exogenous, with the environment representing the natural reservoir for transmission to humans and animals, including livestock and pets. Acquisition mainly occurs through inhalation or secondary to contaminated puncture wounds, including animal bites, leading to respiratory and cutaneous forms, respectively. Bovine mastitis caused by *Nocardia* spp., a form of nocardiosis unique to this species, is acquired through the teat canal. Respiratory involvement is the most prevalent form of nocardiosis in people and hematogenous spread may follow to several organs leading to the systemic form [2]. In cases of pulmonary nocardiosis, metastatic brain abscesses develop in about a third of patients [3]. Cutaneous-subcutaneous nocardiosis is characterised by cellulitis, panniculitis, single or often multiple pyogranulomas and/or abscesses with draining tracts that can ulcerate; involvement of surrounding soft tissues and bone may result in systemic infection. In small animals, recent reports show that the cutaneous-subcutaneous form is becoming the predominant particularly in dogs, compared to the respiratory presentation [4,5,6]. Disseminated disease arising from the lungs or skin seems more frequent in pets than in humans [7].

*Nocardia asteroides* complex are among the most pathogenic and commonly implicated organisms within the genus, although several species can produce clinical infections. Molecular identification of nocardiae has a major role in defining the most effective antimicrobial treatments; the complex has been reorganised into different species (*N. asteroides* types I to VI) based on their antimicrobial susceptibility pattern and molecular characterisation [8,9].

Human nocardiosis is a primarily opportunistic disease with severe clinical presentations in specific risk groups, where progression often leads to disseminated infections; nevertheless, life-threatening infections among immunocompetent humans are also reported [10,11]. Conversely, a primary pathogenic role has been proposed for canine nocardiosis where underlying causes are often unidentified [12]. However, putative predisposing factors have recently emerged, particularly co-infection with canine distemper virus [13,14,15,16] and treatment with immunosuppressants for the management of canine atopic dermatitis [17,18,19].

*N. farcinica* (*N. asteroides* type V) was first isolated in 1888 from a case of bovine farcy [20] and it is classically associated with bovine nocardiosis. This species is known for its predilection for causing life-threatening systemic infection in humans and for its marked degree of intrinsic antimicrobial resistance, including ampicillin, third-generation cephalosporins and aminoglycosides, except amikacin [3,21]. In people, *N. farcinica* is the species most commonly affecting the brain, either as meningitis or brain abscess [22]. The clinical presentation of both systemic and nervous nocardial infections in dogs was reportedly similar to that of humans, for which dogs may represent a suitable model [23].

The present report describes the clinical, pathological, microbiological and molecular characterisation of disseminated nocardiosis in two 14-week-old Dogue de Bordeaux siblings caused by *N. farcinica*.

## 2. Case Presentation

### 2.1. Clinical History and Presentation

Two 14-week-old intact female Dogue de Bordeaux siblings from a litter of nine, presented to a private practice in the UK for evaluation of diarrhoea and lameness of a few days’ duration. Vaccination history was unknown. Both subjects were underweight and showed stunted growth. Dog 1 was 7/10 lame on the right hind limb where a painful swelling was present over the medial metatarsal area. Pyrexia was detected upon physical exam without further abnormalities. Probiotics, gastrointestinal diet, anti-inflammatories (paracetamol, 10 mg/kg PO BID) and antibiotic (amoxicillin-clavulanic acid, 10 mg/kg PO BID) were prescribed for a one-week course. The gastrointestinal disturbance rapidly resolved but improvement of the limb pain and lameness was limited. Radiology indicated a fracture of the IV metatarsal bone of the right hind limb and conservative therapy consisting of cage rest and analgesia (paracetamol BID and buprenorphine at 0.02 mg/kg PO TID) was elected. However, the dog deteriorated significantly in the following days as the limb swelling continued to increase in size before bursting out purulent discharge. The patient was hospitalised for management of the open wound and limb pain but was, unfortunately, euthanised on welfare grounds owing to rapid escalation to laboured breathing and neurological signs (ataxia, seizures and coma).

Dog 2 displayed comparable gastrointestinal signs and was initially treated (excluding analgesia) and responded as dog 1. Shortly after, a painful subcutaneous lump was noted over the left interscapular region, that coincided with onset of 7/10 grade lameness of the left thoracic limb. A reaction to the subcutaneous antibiotic injection administered during the previous consult was suspected and managed with cold compresses and oral paracetamol BID; nevertheless, dog 2 quickly developed non-weight bearing lameness. Radiology detected no abnormalities but observation of a puncture wound compatible with dog bite marks over the shoulder swelling was made while under sedation, alongside another wound detected on the lateral thorax. In-house fine needle aspirate of the shoulder swelling was consistent with an abscess. Cage rest and pain management were prescribed as per dog 1 in addition to amoxicillin at 10 mg/kg IV BID owing to unsettling body temperature. The abscess was lanced and flushed; however, dog 2 continued to worsen with uncontrollable shoulder pain while maintaining analgesia with paracetamol, meloxicam and buprenorphine. Sadly, a rapid progression to neurological signs warranted euthanasia of dog 2, performed the day after that of dog 1.

### 2.2. Pathology and Immunohistochemistry

Following a brief period of freezing and thawing, post-mortem examination was performed 5 (dog 2)–6 (dog 1) days after death. Tissue samples of lungs, liver, spleen, lymph nodes, tonsils, bone marrow, pancreas, brain, peripheral nervous system (sciatic nerves and brachial plexi), stomach, small and large intestines, heart, pericardium, kidneys, urinary bladder, skin, subcutis and muscle were fixed in 10% neutral formalin and processed routinely, embedded in paraffin and sectioned at 4–5 μm and stained with haematoxylin and eosin (H&E), Gram and Ziehl-Neelsen (ZN) stains. Immunohistochemistry (IHC) was performed to investigate the presence of canine distemper and parvovirus infections in tonsil, lung, jejunum, spleen and bone marrow specimens. For this, the protocols below were performed using Dako Autostainer Link 48, with instructions and reagents provided (Dako) unless otherwise stated. Antigen retrieval was achieved by the High Target Retrieval System (TRS) during which slides were heated to 96 °C, cooled to 65 °C, washed with either High pH or Low pH buffers (High: Tris/EDTA buffer pH 9.0 (K8004); Low: Citrate buffer pH 6.1 (K8005)); this also achieved deparaffinization and dehydration. Endogenous peroxidase was blocked using 300 μL EnVision™(Dako; Agilent Technologies, Inc., Santa Clara, CA, USA) FLEX Peroxidase-Blocking Reagent (SM801) for 5 min. Primary antibodies were applied in 300 μL for 20 min, before rinsing with buffer (EnVision™ FLEX Buffer, K8007). For antigen retrieval, slides were incubated with 300 μL EnVision™ FLEX/HRP-labelled polymer for 20 min, followed by two rinses with the same buffer. Tissue sections were incubated with EnVision™ FLEX 3,3′-Diaminobenzidine (DAB) + Substrate Chromogen System for 2 × 5 min in 300 μL for each incubation. Another rinse with buffer followed before 5 min incubation with 300 μL EnVision™ FLEX Haematoxylin counterstain. Three final rinses with distilled water, buffer and distilled water again, respectively, were completed before mounting with DPX. The positive controls included in the IHC assays for both viral antigens consisted of tissue from previous cases of canine distemper and canine parvovirus which were submitted to the diagnostic service with typical gross and histologic lesions.

Upon gross pathology, both carcases were in poor body condition. The most prominent gross changes in dog 1 consisted with a 20 mm long defect of the skin over the dorsal aspect of the right hind foot (right proximal phalangeal region), overlying digits 3 and 4 visible on external examination. A purulent exudate was present in and around the wound. The underlying subcutis was necrotic, purulent and oedematous. The area affected was approximately 40 × 20 mm and extended proximally from the skin wound. There was a complete fracture of the distal end of IV metatarsal bone of the right hindlimb. Lesions present on examination of internal organs included a 40 × 20 mm area of abscessation in the right lung, affecting the caudal aspect of the caudal lobe (Figure 1a). Deforming the capsular profile, both kidneys presented discrete cortical abscesses (Figure 1b). Multifocal, soft yellow areas of malacia and cavitating abscesses surrounded by thin rim of hyperaemia were present in the brain, across cerebral hemispheres and brainstem (Figure 1c). There was marked enlargement of the right popliteal lymph node and mesenteric lymph nodes with suppurative lymphadenitis present in the bronchial lymph nodes.

On external examination of dog 2, there was an 80 mm diameter abscess filled with purulent material on the dorsal neck cranial to the scapula. In the underlying subcutis there was a 40 × 80 mm area of necrotising and purulent tissue bordered by erythema. A second skin wound was present on the mid-thorax. Further areas of abscessation, necrosis and haemorrhage were found in the right kidney, brain and lungs. In the right kidney, the lesions were multifocal and focused on the cortex (Figure 1e). The brain contained a single, 10 mm diameter, demarcated area of haemorrhagic malacia of the left parietal cortex (Figure 1f), in addition to few small areas of mild softening and reddening. The lungs were affected by disseminated, white nodules up to 3 mm in diameter. There was enlargement of the mesenteric, thoracic and other peripheral lymph nodes. The bronchial lymph nodes were markedly enlarged and effaced by suppurative lymphadenitis (Figure 1d). Overall, the main gross differential diagnosis in our cases was considered disseminated *Streptococcus canis* infection.

Microscopically, the histological appearance of lesions was comparable between the two subjects. Examination of the kidney, brain, lung and skin wounds revealed similar changes characterised by areas of necrosis with infiltrating viable and non-viable neutrophils with fewer macrophages and lymphocytes (Figure 2a–c). In the lungs, smaller inflammatory foci were often centred around blood vessels with necrosis of the tunica intima and media and luminal thrombosis, suggesting haematogenous spread of bacteria or endotoxaemia (Figure 2d). Fibrinoid necrosis of veins in sections of the urinary bladder was also detected. Ziehl-Neelsen and Gram stains failed to demonstrate bacteria histologically within any lesions. Immunohistochemistry was negative for canine distemper virus and parvovirus antigens in both dogs’ tissues. 

### 2.3. Bacterial Culture, AST and 16S rRNA Sequencing

Small blocks of post-mortem tissues of lung, liver, kidney, bone marrow and small intestine from both dogs alongside additional bronchial and popliteal lymph nodes from subject 1 and scapular abscess fluid and axillary lymph node from subject 2 were submitted for routine bacteriology. Bacterial culture was performed by first sealing the surface of the tissue specimens with a hot spatula followed by an incision with a sterile scalpel and transferring a small amount of the deeper tissue onto culture plates. Samples were grown on blood agar (BA) base containing 5.0% defibrinated sheep blood (Oxoid, Basingstoke, UK) and fastidious anaerobes agar (FAA, E&O Laboratories Ltd., Bonnybridge, UK) at 37 °C for five days under aerobic and anaerobic conditions, respectively. Direct smears were obtained for cytology by organ impression and Gram stained, with the exception of bone marrow and intestine. Numerous filamentous Gram-positive organisms approximately 0.5–1 µm in diameter and up to 25 µm in length showing branching behaviour, metachromatic staining with beaded appearance were observed in smears from the kidney in both subjects. The same was seen in the lung and bronchial node of dog 1 and the scapular abscess and axillary node of dog 2. Gram-negative rods were additionally detected in the abscess smear from dog 2. Acid-fast stain (modified Ziehl-Neelsen—MZN) was performed for the presumptive differentiation between *Nocardia* spp. and other filamentous Gram-positive organisms such as *Actinomyces* spp., which indicated a partially acid-fast actinomycetes (Figure 3). Accordingly, culture on Sabouraud dextrose agar (SDA) (Oxoid, Basingstoke, UK) was set up. Pure moderate growths of chalky-white, dry and powdery bacterial colonies were identified on BA inoculated from kidney (both dogs), lung and bronchial node (dog 1) and axillary node (dog 2) after 48 h; the abscess fluid (dog 2) yielded the same in mixed growths with a non-haemolytic coliform and scanty yeasts. Moderate growths of powdery colonies were also identified on SDA at 48–72 h from the corresponding specimens. No bacterial growth was obtained from samples of liver, spleen and bone marrow from both dogs and from the popliteal node of subject 1 and lung of subject 2 (Table 1). Faecal coliforms and enterococci were isolated from small intestinal samples consistent with commensal flora. Organism identification was achieved by MALDI-TOF MS (Matrix-Assisted Laser Desorption/Ionization Time-of-Flight Mass Spectrometry; Bruker Daltonics, Bremen, Germany) with a score >2.2, which identified the main cultured organism as *Nocardia farcinica* and the abscess coliform and yeast as *Escherichia coli* and *Candida albicans*, respectively. To complete the phenotypic characterisation of the isolates, antimicrobial susceptibility testing (AST) was performed by broth microdilution using a small animal susceptibility panel (COMPGP1F, TREK Diagnostic System, West Sussex, UK) and results interpreted according to the Clinical and Laboratory Standards Institute (CLSI) criteria for human isolates [24]. 16S rRNA PCR amplification was performed for species confirmation according to Randall et al., 2015 [25]. PCR products were resolved by electrophoresis through a 1.5% (*w*/*v*) TAE agarose gel. DNA was purified using NucleoSpin^®^ Gel and PCR Clean-up (Takara Bio Inc., Kusatsu, Japan) and its concentration and quality assessed by NanoDrop microvolume system [26] prior to sequencing (Source BioScience, Nottingham, UK). Reverse and complement sequences and chromatograms were edited and aligned using ChromasPro software (Technelysium Pty Ltd., South Brisbane, Australia) and nucleotide BLAST search was conducted through the NCBI website.

## 3. Discussion

Nocardiosis is an emerging disease among humans and animals worldwide with increasing incidence in non-tropical areas. Nocardial infections in dogs tend to be severe and difficult to treat, often resulting in death or euthanasia [27]. The lung and skin are considered primary sites of involvement leading to the respiratory and cutaneous/subcutaneous forms, respectively. For the present cases, symptoms were largely non-specific as disseminated forms may present with variable manifestations depending on the affected sites. Nocardiosis remains a rare and neglected disease in people and dogs and, unless suspected, diagnosis can be easily missed [28]. Besides, disseminated forms are most frequently underreported [29]. Indeed, reports of systemic nocardiosis in dogs manifesting on initial presentation as lameness are infrequent [17,30]. The cutaneous injuries are here regarded as the portals of entry in consideration of an infected open wound co-localizing with the digit fracture in dog 1 and of the subcutaneous abscess over the bite wound in dog 2. Cytological evidence of predominant nocardiae in the abscess direct smear of dog 2 may be supportive, although no cutaneous specimen was submitted for dog 1. Giving that nocardiae are often isolated from clinical specimens in mixed culture and may be discarded as contaminants, cytology should be used to interpret culture findings. Higher prevalence of nocardiosis in roaming or fighting animals where lesions are associated with scratches or bite wounds have been previously reported [5]. Systemic spread is thought to have originated from the cutaneous lesions; however, penetration through the respiratory route cannot be excluded. Whether the dogs were simultaneously exposed to the same environmental source of *Nocardia* spp. or, whether an index case transmitted to the other is anyhow more challenging to demonstrate. Interestingly, transmission of *Nocardia* spp. do not tend to occur from person to person, and outbreaks are rare. Since the 80s, a handful of hospital outbreaks have been reported in immunosuppressed humans [31]. Therefore, to the best of the authors’ knowledge, this represents the first documented small-scale outbreak of *Nocardia* spp. among dogs.

No immunosuppression or other predisposing factor could be ascertained for these cases beyond their negative immunohistochemical test results for canine distemper and parvovirus. On the other hand, the age of the present subjects is consistent with literature suggesting that *Nocardia* spp. is a primary pathogen for young dogs, who develop disseminated forms more often than adults [12,23]. Our cases of systemic nocardiosis display gross similarities with previously reported disseminated infections sustained by other *Nocardia* species in dogs, such as *N. cyriacigeorgica* [12] and *N. veterana* [15]. Gram-stained histological sections of the brain, lung, skin and renal lesions failed to reveal bacterial organisms in both dogs. Inferior visualisation of *Nocardia* spp. on Gram-stained preparations compared to silver stains has been described [12]. Gomori’s methenamine silver nitrate (GMS) and Brown and Brenn’s methods [32] have proven valuable for the presumptive identification of actinomycetes in biopsied tissues. These filamentous organisms could sometimes be mistaken for fungal structures on histology [18] and Periodic Acid Schiff (PAS) stain aids distinction from their PAS-positive fungal mimickers [32]. Definitive diagnosis is accomplished by bacterial culture and identification. Colonies obtained aerobically after 48–72 h on conventional laboratory media have characteristic morphology; however, prolonged incubation may encourage formation of pronounced aerial filaments and pigmentation that warrant differentiation from fungi. Gram-stained direct microscopy of colonies is essential for their differentiation. *Actinomyces* spp. constitutes the main differential diagnosis in the microbiology laboratory as could *Streptomyces* spp. However, whilst the former is also an important clinical differential in cases of canine cutaneous and respiratory nocardiosis, the latter is an extremely infrequent pathogen, seldom associated with human mycetoma [33]. Using a modified acid-fast stain such as MZN or Fite-Faraco is an excellent method for distinguishing between *Nocardia* and *Actinomyces* or *Streptomyces* species.

MALDI-TOF MS has been reported as an adequate tool for identification of *Nocardia* spp. isolates [34], including that of veterinary origin [35]. Some challenges are posed by the continuously evolving taxonomy and large number of species (currently >100), with uncommon nocardiae underrepresented in the available databases [36]. It must be highlighted that MALDI-TOF MS database does not include *Nocardia kroppenstedtii* to which *N. farcinica* is closely related. For these reasons, sequencing of the 16S rRNA, *hsp65* and *secA1* genes [37] are paramount for unambiguous species characterisation. The present isolates were confirmed to be *N. farcinica* by 16S rRNA sequencing with 99.8% identity.

Regarding the antimicrobial susceptibility and therapeutic management of nocardiosis, infections are commonly refractory to conventional antimicrobial therapy and require prolonged antibiotic administration. Trimethoprim-sulfonamides are the drugs of choice for treatment of both human [38] and canine [5] infections, including central nervous system infections [10]. Additional antimicrobials can be used alone or in combination with potentiated sulphonamides, including amikacin, linezolid, moxifloxacin and late generation beta-lactams (cefotaxime, ceftriaxone and imipenem-cilastatin) [39]. Multidrug resistant *Nocardia* spp. isolates, including *N. farcinica*, have been identified among human [40,41], bovine [42] and canine isolates [15]. MIC findings from the present isolates showed the organism was susceptible to trimethoprim-sulfamethoxazole, pradofloxacin and amikacin whilst high MICs were recorded for beta-lactams including imipenem and tetracyclines (Table 2). Reports of successful treatment of nocardiosis in dogs include mainly that of cutaneous infections [17,43] and, less commonly, of other forms [30,44].

No specific preventative or control procedures are available for nocardiosis in pets; hygienic environmental conditions, early diagnosis, adequate initial antimicrobial therapy based on culture and AST and control of potential predisposing factors such as immunosuppressive conditions may reduce the severity and mortality rates of infection.

## 4. Conclusions

In conclusion, fatal *Nocardia farcinica* systemic infection is described here in two puppy siblings representing a small-scale outbreak within a canine breeding establishment in the UK. Due to low prevalence and non-specific clinical manifestations, disseminated nocardiosis in dogs is usually misdiagnosed whilst the prognosis and efficacy of therapy may be uncertain even when a diagnosis is reached ante-mortem. Further research into the virulence, clinical-epidemiological features, predisposing factors and type characterisation of *Nocardia* spp. strains isolated from human, animal and environmental origins may help understand the risk factors and improving the patient management relative to nocardial infections.

## Figures and Tables

**Figure 1 vetsci-10-00028-f001:**
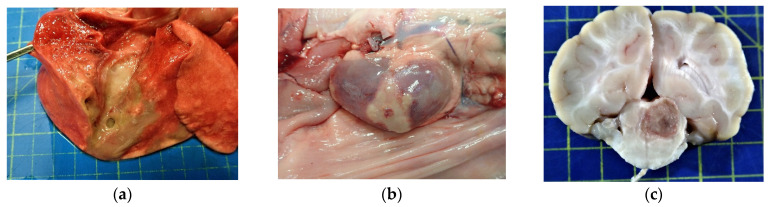
Dog 1 (top): Lung (**a**), a focal, poorly demarcated area of suppurative exudate was present in the right caudal lung lobe; Kidney (**b**), a large, necrosuppurative focus in the cortex of the right kidney is visible from the capsular surface; Brain (**c**), a focal area of malacia and haemorrhage replacing the right rostral colliculus. Dog 2 (bottom): Lymph node (**d**), an enlarged bronchial lymph node showing diffuse necrotising and suppurative lymphadenitis on the cut surface; Kidney (**e**), multifocal necrosuppurative lesions were present in the cortices of the kidneys; Brain (**f**), a focal area of malacia and haemorrhage in the left parietal cortex.

**Figure 2 vetsci-10-00028-f002:**
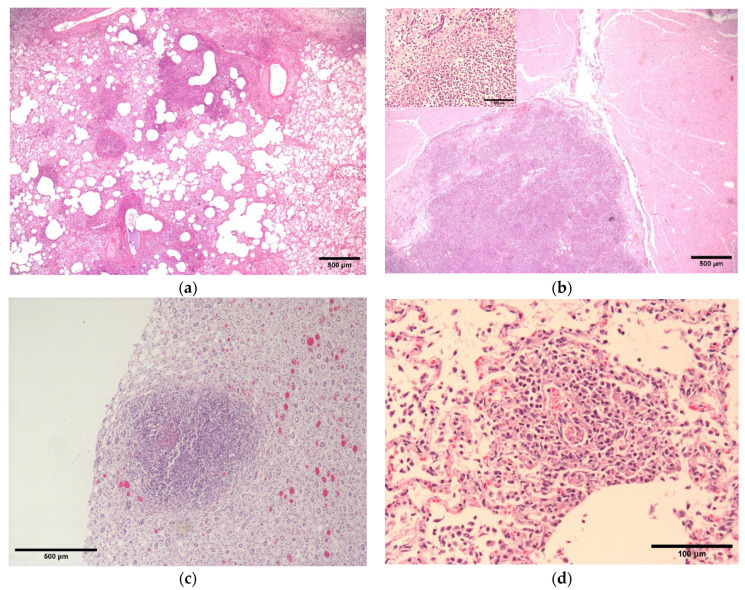
Dog 1 (top): Lung (**a**), multifocal areas of neutrophilic inflammation which often appeared centred on blood vessels, extending into and effacing the alveoli; Brain (**b**), lesions in the brain were characterised by replacement of neuroparenchyma by necrotic debris mixed with neutrophils and fewer gitter cells. The inset shows a close up of the inflammatory cells. Dog 2 (bottom); Kidney (**c**), Discrete large focus of pyogranulomatous inflammation, medulla; Lung (**d**), a focus of suppurative vasculitis with thrombosis. All HE stains.

**Figure 3 vetsci-10-00028-f003:**
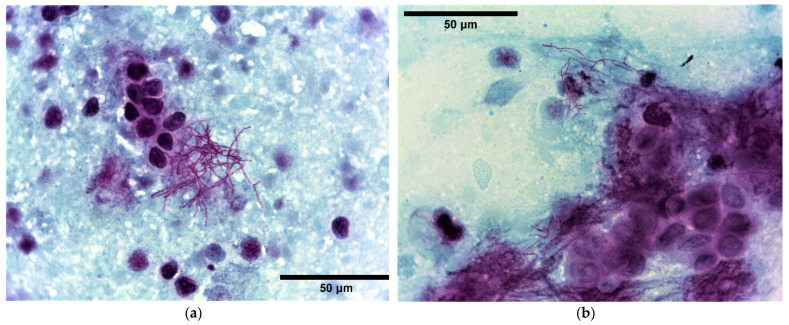
Dog 1, Lung (**a**) and Dog 2, kidney (**b**): Impression smears of canine lung and kidney abscesses, Modified Ziehl-Neelsen (MZN) stain. Note the MZN-positive branching filamentous organisms. ×100 objective.

**Table 1 vetsci-10-00028-t001:** *Nocardia* spp. detection by direct smear and bacterial culture of post-mortem tissue specimens.

	Tissue Type (Including Abscess)	MZN-Positive	Nocardia Culture Positive
*Dog 1*	Lung	Yes	Yes
	Liver	No	No
	Kidney	Yes	Yes
	Spleen	No	No
	Bone Marrow	No	No
	Small Intestine	No	No
	Bronchial lymph node	Yes	Yes
	R Popliteal lymph node	No	No
*Dog 2*	Lung	No	No
	Liver	No	No
	Kidney	Yes	Yes
	Spleen	No	No
	Bone Marrow	No	No
	Small Intestine	No	No
	L Scapular abscess fluid	Yes	Yes
	L Axillary lymph node	Yes	Yes

**Table 2 vetsci-10-00028-t002:** Antimicrobial susceptibility testing of *Nocardia farcinica* by broth microdilution using a veterinary panel (COMPGP1F Vet AST Plate). Interpretation of results was conducted using human clinical breakpoints according to CLSI 2011 [24].

Antibiotic	*Broth Microdilution*
Ampicillin	>8 (R)
Amoxicillin-clavulanic acid	8 (R)
Cefovecin	>8 (R)
Cefpodoxime	>8 (R)
Imipenem	>4 (R)
Amikacin	<16 (S)
Gentamicin	>16 (R)
Doxycycline	>0.5 (R)
Tetracycline	>1 (R)
Minocycline	2 (R)
Marbofloxacin	4 (R)
Pradofloxacin	≤0.25 (S)
Trimethoprim-sulfamethoxazole	≤2 (S)
Nitrofurantoin	>64 (R)
Rifampicin	>2 (R)
Vancomycin	>16 (R)

## Data Availability

Not applicable.

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
