# Peer review of "Disseminated Nocardiosis Caused by *Nocardia farcinica* in Two Puppy Siblings"

_vetsci, 2022, doi:10.3390/vetsci10010028_

Round 1

Reviewer 1 Report

The case study presented the clinical history, microbiologic findings (and molecular characterization), cytological, gross and histopathological findings of  disseminated nocardiosis due to Nocardia farcinica in two 14 week-old Dogue du Bordeaux sibling pups. 

The methods used in the study were clearly stated and reproducible. There is clear outline of the histological processing methods, commercial IHC stains for canine distemper virus and canine parvovirus. The microbiological culture methods,  nucleic acid extraction and sequence analysis (completed using the BLAST program, available at http://www.ncbi.nlm.nih.gov/BLAST) were thoroughly described and could be reproduced.

There were no statistical analyses used in this study.

The references used in the paper are relevant and where appropriate, relatively recent (within the past 5-7 years).

This is the first case study of a small outbreak of Nocardia farcinica in canines.The researchers’ work is significant and original.  The manuscript is clearly written, technically sound and scientifically valid. In my opinion, the manuscript is suitable for publication after some minor suggested changes (listed below).

Suggested minor corrections

Line 183 to 184

Could the authors please include a sentence stating the positive controls used for a) canine parvovirus antigen IHC and b) canine distemper antigen IHC

Figure 2

Please include scale bars on all photomicrographs

Line 198

“2.3. Bacterial culture, AST and 16R rRNA sequencing”  should be “2.3. Bacterial culture, AST and 16S rRNA sequencing” 

Comments:

Should be 16S rRNA

Line 234

16S rRNA PCR amplification was 234 performed for species confirmation according to [21]. 

Comment: Please complete the sentence

Author Response

Response to the Reviewers of the submitted Case Report (Manuscript ID vetsci-2106961):

Disseminated nocardiosis caused by Nocardia farcinica in two puppy siblings

Dear reviewers, many thanks for working on our submitted manuscript and for your kind feedback and useful corrections. Please find below our response where we tried to address your comments one by one. We have been pleased to receive your insightful suggestions and made changes accordingly in the re-submitted manuscript in track-changes mode (file named vetsci-2106961_FZ) or answered your points below in the present document. We remain available to receive further feedback and work on it, might this be needed afterwards.

Reviewer 1

Point 1: Line 183 to 184

Could the authors please include a sentence stating the positive controls used for a) canine parvovirus antigen IHC and b) canine distemper antigen IHC

Response 1: Many thanks for addressing this point; mention of the positive controls used for canine parvovirus and canine distemper IHC antigens have now been added to the manuscript at lines 147-150 (section describing the methodology). Positive controls used by the pathologists consisted of tissue from known cases of canine parvovirus and canine distemper virus submitted to our diagnostic service which presented typical gross and histologic lesions. I would love to say that the positive controls are always diagnosed with PCR as well, however this is not always the case as we often rely on previous test results conducted ante-mortem which may include molecular detection, point-of-care testing and/or serology.

Point 2: Figure 2

Please include scale bars on all photomicrographs

Response 2: We apologize for not having included the scale bars on all original photomicrographs submitted; we have now re-submitted the zipped files for the Figures 2a to 2d (histology) and also 3a-3b (cytology) that include the scale bar. All these figures have been replaced also in the Figure 2 & 3 panels in the re-submitted manuscript.

Point 3: Line 198

“2.3. Bacterial culture, AST and 16R rRNA sequencing” should be “2.3. Bacterial culture, AST and 16S rRNA sequencing” 

Comments:

Should be 16S rRNA

Response 3: Many thanks for spotting this out, it has been corrected into 16S rRNA.

Point 4: Line 234

16S rRNA PCR amplification was 234 performed for species confirmation according to [21]. 

Comment: Please complete the sentence

Response 4: Many thanks for this input, the sentence has included complete mention of the authors referenced for the PCR methodology in the square brackets at line 235.

Reviewer 2 Report

Dear Authors,

I return only my compliments for the work done very well and for its presentation as a Case Report.

I have no other comments except to thank Scholars like you who report neglected or even underestimated pathologies and/or infections, allowing the Scientific Community to broaden the horizons of infectious diseases from a One Health perspective.

Author Response

Dear Reviewer,

Many thanks for your kind and encouraging feedback on our submitted Case Report, we are very pleased with your response and wanted to thank you very much.

Kindest,

The Authors

Reviewer 3 Report

Flavia and colleagues (vetsci-2106961) presented a typical clinical investigation of cases associated with infection by Nocardia farcinica, the overall study is interesting with well-supported data and elegant writings. A few more questions remains.

1. During the bacterial culture, how many culture conditions and medium were used, for those colonies on the plate, did you conduct the MALDI-TOF analysis for all colonies, I am curious if other type of bacteria were also characterized, this is usually the case in clinics, not only one type of bacteria is there.

2. For the identification of Nocardia farcinica, I would further conduct biochemical tests for this particular species, all whole genome sequencing would be a big plus.

3. Figure 1. I would suggest to presents all the same tissues or organs for two patients, side by side, however, the lung and kidney samples were mixed here.

4. I am wondering where this Nocardia farcinica is from, have you conducted this survey on the feed or the environment or the skin of the patient? 

Finally, I would suggest adding more recent publications (within five years) within the field for this particular bacteria. 

Author Response

Response to the Reviewers of the submitted Case Report (Manuscript ID vetsci-2106961):

Disseminated nocardiosis caused by Nocardia farcinica in two puppy siblings

Dear reviewers, many thanks for working on our submitted manuscript and for your kind feedback and useful corrections. Please find below our response where we tried to address your comments one by one. We have been pleased to receive your insightful suggestions and made changes accordingly in the re-submitted manuscript in track-changes mode (file named vetsci-2106961_FZ) or answered your points below in the present document. We remain available to receive further feedback and work on it, might this be needed afterwards.

Reviewer 3

Point 1. During the bacterial culture, how many culture conditions and medium were used, for those colonies on the plate, did you conduct the MALDI-TOF analysis for all colonies, I am curious if other type of bacteria were also characterized, this is usually the case in clinics, not only one type of bacteria is there.

Response 1: With reference to the culture conditions and media used for the isolation of the N. farcinica isolate from post-mortem samples, we conducted routine plating on 5.0% defibrinated sheep blood agar (BA) and FAA incubated at 37°C under aerobic and anaerobic conditions, respectively. N. farcinica was isolated in pure growths from all bacterial culture-positive samples incubated aerobically except the intestinal contents (yielding coliforms and enterococci consistent with normal commensal gut flora) and the interscapular abscess fluid (dog 2), from where it was isolated in mixed, predominant growths alongside lesser Escherichia coli colonies and scanty Candida spp. yeasts. Multiple colonies of N. farcinica from each positive organ’s culture plate were subjected to MALDI-TOF identification from primary cultures as well as from subculture of single colonies to evaluate purity of the isolated microorganisms. The identity of the coliform and yeast isolated from the abscess fluid were also confirmed in their identity following subculture. With the exception of the abscess fluid sample, pure moderate growths of N. farcinica were confirmed in the other positive specimens, which may reflect the true pathogenic dissemination to internal sites. On the other hand, cutaneous-subcutaneous abscess material may typically contain other types of bacteria, as abscesses are often polymicrobial in nature and opportunistic bacteria or contaminating commensal flora are commonly isolated from this sample type. We believe this was nicely captured also by the cytological findings showing a single type of filamentous bacteria from the internal organs’ impression smears in both dogs whilst gram-negative bacilli were seen in low numbers in the abscess fluid cytology together with filamentous gram-positives. Sampling technique of post-mortem tissues plays a crucial role both in the post-mortem room and microbiology laboratory as specimens may often become contaminated; nevertheless, a good technique may help reducing the chances of external contamination and reveal a single agent as most likely associated to the presenting clinical condition.

Point 2. For the identification of Nocardia farcinica, I would further conduct biochemical tests for this particular species, all whole genome sequencing would be a big plus.

Response 2: We agree with the reviewer that both biochemical identification and WGS data would broaden knowledge regarding this pathogen. However, since we introduced MALDI-TOF identification in our laboratory, we have discontinued the biochemical testing, as MALDI-TOF results appear to be more reliable (https://sfamjournals.onlinelibrary.wiley.com/doi/full/10.1111/lam.12526). WGS analysis would be very useful for establishing a clinical diagnosis when bacterial culture fails, which was not the case in this study. In addition, the funding for this project was limited and we could not afford the costs for WGS. Nevertheless, we would be happy to collaborate and share the isolates with other groups who may have an interest in studying Nocardia spp. and could include these isolates in WGS analysis, therefore adding to the current knowledge on this pathogen.

Point 3. Figure 1. I would suggest to presents all the same tissues or organs for two patients, side by side, however, the lung and kidney samples were mixed here.

Response 3: Dear reviewer, many thanks for this suggestion - we agree that the Figure 1 panel would be nicer. We have now submitted two novel figures consisting of the kidney of dog 1 (Figure 1b) and of the bronchial lymph node of dog 2 (Figure 1d) to complete the panel of gross images. The new Figure 1 panel arrangement is now showing the same organs for the two patients side by side with dog 1 at the top and dog 2 at the bottom, respectively. The panel is therefore now composed of six gross images disposed on two rows of three figures each, that have been renamed from Figure 1a to Figure 1f. Changes are integrated in the re-submitted manuscript, both in the text (at lines 159, 162,169, 170, 171 and 174) and Figure 1 panel and itself where captions were added for the novel images.

Point 4. I am wondering where this Nocardia farcinica is from, have you conducted this survey on the feed or the environment or the skin of the patient? 

Response 4: Unfortunately, an environmental surveillance investigation was not conducted as part of this study owing to funding constraints. We also wondered what the source of this organism was in this case, however being Nocardia spp. opportunistic bacteria widely found in the environment, identifying the exact source could have involved many resources with potentially little outcome. Having had appropriate resources, we believe it would have been relevant sampling the animal feed first as this has been previously incriminated to cases of tuberculosis in small animals fed with raw and commercial pet food diets, which, regrettably, we could not carry out. With regard to point 2 above (from reviewer 2), WGS would be extremely useful to investigate the molecular epidemiology and potential sources of this isolate. Based only on 16S sequence identity, our isolate displayed closest similarities to N. farcinica strains isolated from both plant material and human clinical samples (16S sequence percentage similarity range between 99.71 and 99.78 %).

Point 5: Finally, I would suggest adding more recent publications (within five years) within the field for this particular bacteria. 

Response 5: Dear reviewer, thanks for this suggestion; some more recent publications have been now referenced which are relevant to this particular bacterial genus, particularly in the introduction where more historical references where included. These are found at lines 56 (Reference 2), 62 (Reference 6), 77 (Reference 16), 84 (Reference 22) and 308 (Reference 35).